# Navigable Graphs for High-Dimensional Nearest Neighbor Search: Constructions and Limits

**Haya Diwan**
New York University
hd2371@nyu.edu

**Jinrui Gou**
New York University
jg6226@nyu.edu

**Cameron Musco**
UMass Amherst
cmusco@cs.umass.edu

**Christopher Musco**
New York University
cmusco@nyu.edu

**Torsten Suel**
New York University
torsten.suel@nyu.edu

## Abstract

There has been recent interest in *graph-based nearest neighbor search methods*, many of which are centered on the construction of (approximately) *navigable* graphs over high-dimensional point sets. A graph is navigable if we can successfully move from any starting node to any target node using a greedy routing strategy where we always move to the neighbor that is closest to the destination according to the given distance function. The complete graph is obviously navigable for any point set, but the important question for applications is if sparser graphs can be constructed. While this question is fairly well understood in low-dimensions, we establish some of the first upper and lower bounds for high-dimensional point sets. First, we give a simple and efficient way to construct a navigable graph with average degree $O(\sqrt{n \log n})$ for any set of $n$ points, in any dimension, for any distance function. We compliment this result with a nearly matching lower bound: even under the Euclidean metric in $O(\log n)$ dimensions, a random point set has no navigable graph with average degree $O(n^\alpha)$ for any $\alpha < 1/2$. Our lower bound relies on sharp anti-concentration bounds for binomial random variables, which we use to show that the near-neighborhoods of a set of random points do not overlap significantly, forcing any navigable graph to have many edges.

## 1 Introduction

The concept of a *navigable graph* has arisen repeatedly over the decades, perhaps most famously in Kleinberg's work on understanding Milgram's "Small World" experiments from the 1960s [Kleinberg, 2000a,b, Milgram, 1967]. Concretely, suppose we are given $n$ points $x_1, \ldots, x_n \in \mathcal{X}$ in some input domain $\mathcal{X}$, a distance function $D : \mathcal{X} \times \mathcal{X} \to \mathbb{R}^{\geq 0}$, and a directed graph $G = (V, E)$, where each vertex in $V = \{1, \ldots, n\}$ is associated with one of our points. $G$ is said to be *navigable* if the standard *greedy routing* algorithm successfully finds a path between any starting vertex $s \in V$ and any target vertex $t \in V$.[1] In particular, letting $\mathcal{N}(s)$ denote the out-neighbors of $s$, this algorithm first navigates to $r \in \mathcal{N}(s)$ which minimizes $D(x_r, x_t)$. At each subsequent step, we navigate to the out-neighbor of the current node that minimizes the distance to $x_t$, terminating once we reach $x_t$, or if no neighbor has an associated point that is closer to $x_t$ than the current node.

It has been observed that many real-world networks (the internet, airport networks, social networks, etc.) are either navigable or almost navigable, where $x_i$ plays the role of, e.g., the physical coordinates of a server or individual and $D$ is the standard Euclidean distance or some other metric [Boguñá et al.,

---

[1] $G$ is further considered "small-world" if greedy routing always terminates in a small number of steps.

38th Conference on Neural Information Processing Systems (NeurIPS 2024).

2009]. Moreover, there has been interest in showing that natural generative models for networks produce navigable graphs with good probability [Kleinberg, 2000a, Watts and Strogatz, 1998].

## 1.1 Constructing Sparse Navigable Graphs

More recently, significant work has studied the problem of *constructing* navigable or "approximately" navigable graphs given a point set $x_1, \ldots, x_n \in \mathcal{X}$ and distance function $D$. For any $D$ and any point set, the complete graph is navigable, so more concretely, the goal is to construct a navigable graph that is *as sparse as possible*. At a high-level, this objective underlies many recently developed graph-based approximate nearest neighbor search methods such as DiskANN [Subramanya et al., 2019], the Hierarchical Navigable Small World (HNSW) method [Malkov and Yashunin, 2020, Malkov et al., 2014], and the Navigating Spreading-out Graph (NSG) method [Fu et al., 2019].[2] Such methods have shown remarkable empirical performance, outperforming state-of-the-art implementations of popular approximate nearest neighbor search algorithms such as product-quantization and locality-sensitive hashing [Johnson et al., 2021, Jégou et al., 2011, Indyk and Motwani, 1998, Andoni et al., 2015]. The computational efficiency of the graph-based methods is governed by the number of edges in the graph being searched, motivating the need for sparse navigable graphs.

Despite this recent interest, there has been relatively little theoretical work on the problem of constructing navigable graphs. When the input points lie in $\mathbb{R}^d$ and $D$ is the Euclidean distance function, it is not hard to check that the Delaunay graph is navigable.[3] While the Delaunay graph has average degree $O(1)$ in dimension $d = 2$ (since it is planar) it can have average degree $O(n)$ in dimension $d = 3$ or higher [Klee, 1980]. A better bound can be obtained via the so-called *sparse neighborhood graphs* of [Arya and Mount, 1993], which are shown to be navigable for any point set in $\mathbb{R}^d$ under the Euclidean distance and have average degree $2^{O(d)}$.[4] While this results in a sparse navigable graph for small values of $d$, the degree bound is no better than that of the complete graph for $d = \Omega(\log n)$. Given that modern applications of nearest neighbor search often involve high dimensional data points, it is natural to ask if anything better can be done in high dimensions.

## 1.2 Our Results

The main contribution of this work is to provide tight upper and lower bounds on the sparsity required to construct navigable graphs for high-dimensional point sets. In particular, we prove two main results. The first is a strong upper bound that follows from a straight-forward graph construction:

**Theorem 1.** *For any input domain $\mathcal{X}$, point set $x_1, \ldots, x_n \in \mathcal{X}$, and distance function $D : \mathcal{X} \times \mathcal{X} \to \mathbb{R}^{\geq 0}$ such that $D(x_i, x_i) = 0$ for all $i$ and $D(x_i, x_j) > 0$ for $x_j \neq x_i$, it is possible to efficiently construct a directed navigable graph with average degree at most $2\sqrt{n \ln n}$. Moreover, the graph has the additional "small world" property: greedy routing always succeeds in at most 2 steps.*

Theorem 1 establishes that, even in arbitrarily high dimension, it is possible to beat the naive complete-graph solution, which has $O(n^2)$ edges (average degree $n$). The result is proven in Section 3. It is based on an simple construction, reminiscent of existing techniques for building nearest neighbor search graphs: we take the union of a $O(\sqrt{n \log n})$-nearest neighbor graph, and a random graph with average degree $O(\sqrt{n \log n})$ [Malkov et al., 2014, Kleinberg, 2000b, Subramanya et al., 2019].

Surprisingly, Theorem 1 holds for essentially any distance function, even if it is not a metric. Moreover, the construction is efficient: the navigable graph can be computed in $O(n^2(T + \log n))$ time, where $T$ is a bound on the cost of computing $D(x_i, x_j)$ for any $i, j$. Given the generality of Theorem 1, we might expect that navigable graphs with even fewer edges could be constructed under additional assumptions – e.g., if we considered only the special case where the input domain is $\mathbb{R}^d$ and $D$ is the Euclidean distance. Our next result rules this out when $d = \Omega(\log n)$. In particular, we show:

**Theorem 2.** *Let $x_1, \ldots, x_n \in \mathbb{R}^d$ be vectors with i.i.d. random $\pm 1$ entries, and let $D(x_i, x_j) = \|x_i - x_j\|_2$ be the Euclidean distance. For any parameter $\delta > 0$, if $d \geq \frac{c}{\delta^2} \log n$ for a fixed constant $c$, then with high probability, any navigable graph for $x_1, \ldots, x_n$ requires average degree $\Omega(n^{1/2-\delta})$.*

---

[2] Note that none of these methods actually claim to construct sparse graphs that are navigable in the worst-case. They use heuristics to build graphs that are "approximately navigable", meaning that, empirically, greedy search tends to find good approximate nearest neighbors when run on the graphs.

[3] The Delaunay graph connects $i, j$ if $x_i$ and $x_j$ have adjacent cells in a Voronoi diagram for $x_1, \ldots, x_n$.

[4] Such graphs are closely related to but not the same as "relative neighborhood graphs" [Toussaint, 1980].

Theorem 2 is proven in Section 4. It is a corollary of our more general Theorem 4, which also implies a lower bound of $\Omega(n^{1/2}/\log n)$ average degree when $d = \Omega(\log^3 n)$. The proof starts with a straight forward observation: in order for greedy routing to make progress towards a destination node $x_i$, any node within the $k$-nearest neighbor set of $x_i$, for any $k$, must include an edge to some other node in that set (possibly $x_i$ itself). Using sharp anti-concentration bounds for binomial random variables [Ahle, 2017, Cramér, 2022], we argue that, when $k = O(\sqrt{n})$ and when $x_1, \ldots, x_n \in \{-1, 1\}^d$ are random for large enough $d$, the nearest neighbor sets for different destination nodes have very small pairwise intersections. Intuitively, they are nearly independent random sets of size $O(\sqrt{n})$, and thus have expected overlap close to 1. This small overlap means that few edges can be used to 'cover' the required connections within different nearest neighborhoods, giving a lower bound on the average degree of any navigable graph.

We remark that Theorem 2 cannot be improved significantly in its bound on the dimension $d$. As mentioned, for the Euclidean distance over $\mathbb{R}^d$, it is possible to construct navigable graphs with $2^{O(d)}$ edges for any $d$ dimensional point set using the sparse neighborhood graphs of [Arya and Mount, 1993]. This leads to average degree less than $n^{1/2}$ when $d = c \log n$ for a small enough constant $c$.

## 1.3 Outlook

Together, Theorems 1 and 2 help complete the picture of what level of sparsity is achievable when constructing navigable graphs in high-dimensions. However, these bounds are not the end of the story. For one, there are many variations on simple greedy search that would lead to other notions of navigability. For example, in nearest neighbor search applications, a version of greedy search called *beam search*, which explores multiple greedy paths, is often preferred [Subramanya et al., 2019].

Beyond the average degree, which we focus on in this work, the *maximum degree* of a navigable graph is also a natural metric, governing the maximum complexity of each iteration of greedy search. Unfortunately, for the navigability problem we study, we show in Section 4.3 that there are point sets for which *every* navigable graph must have maximum degree $n$. It would be interesting if relaxations of the problem or a more flexible search method can avoid this limitation.

Finally, an important direction for future work is to prove end-to-end approximation guarantees for graph-based nearest neighbor search algorithms. Since finding *exact* nearest neighbors in high dimensions suffers from challenges related to the curse of dimensionality, a reasonable goal would be to prove that greedily routing towards any query $q \in \mathcal{X}$ converges on an $\alpha$-approximate nearest neighbor $x_j$ satisfying $D(q, x_j) \leq \alpha \cdot \min_i D(q, x_i)$ for some $\alpha \geq 1$. This is the sort of guarantee that locality sensitive hashing and other methods can achieve with query time that is provably *sublinear in n*, i.e., without needing to directly compare $q$ to all vectors $x_1, \ldots, x_n$ [Kleinberg, 1997, Kushilevitz et al., 1998, Indyk and Motwani, 1998, Har-Peled, 2001]. Importantly, if regular greedy search is applied from an arbitrary starting node, navigability is a *necessary condition* for $\alpha$-approximate nearest-neighbor search. In particular, for any finite $\alpha$, if $q \in \{x_1, \ldots, x_n\}$, $\min_i D(q, x_i) = 0$, so we must return $q$ exactly. However, navigability is not a *sufficient condition* for greedy search to succeed, as it does not guarantee any level of approximation for queries $q$ that are not in $\{x_1, \ldots, x_n\}$. For some initial work on this more challenging problem, we refer to the reader to Laarhoven [2018], Prokhorenkova and Shekhovtsov [2020], and Indyk and Xu [2023].

## 2 Preliminaries

**Notation.** Throughout, we consider a set of $n$ distinct points $x_1, \ldots, x_n \in \mathcal{X}$ for some input domain $\mathcal{X}$ and a distance function $D : \mathcal{X} \times \mathcal{X} \to \mathbb{R}^{\geq 0}$. $D(x_i, x_j)$ denotes the distance from point $j$ to point $i$. We assume only that $D(x_i, x_i) = 0$ for all $i$ and that $D(x_i, x_j) > 0$ for $x_j \neq x_i$.[5] Our main upper bound, Theorem 1, does not require $D$ to be a metric, or even to be symmetric. Our main lower bound, Theorem 2, holds against the standard Euclidean distance $D(x_i, x_j) = \|x_i - x_j\|_2$.

We aim to construct a directed graph $G = (V, E)$, where each vertex in $V = \{1, \ldots, n\}$ is associated with one of our input points. Each edge $e \in E$ is an ordered pair $(i, j)$, indicating that there is an

---

[5]Even these assumptions are not needed, but they make our definition of "navigable" more natural. Otherwise, we must allow for greedy routing to navigate to any vertex $x_j \in \operatorname{argmin}_{x_1, \ldots, x_n} D(x_j, x_t)$ given target $x_t$.

edge from node $i$ to node $j$. Throughout, we let $\ln n$ denote the natural base-$e$ logarithm of $n$, and $\log n$ denote the base-2 logarithm of $n$.

---

**Algorithm 1** Greedy Search

---

1: **Input:** Graph $G$ over nodes $\{1, \ldots, n\}$, starting node $s$, query point $\bar{x}$.
2: **Output:** A point $x_j$ that is ideally close to $\bar{x}$ with respect to distance function $D$.
3: $j \leftarrow s$, Done $\leftarrow$ False
4: **while** Done == False **do**
5:     **if** $\mathcal{N}(j) = \emptyset$ **then**
6:         Done $\leftarrow$ True.
7:     **else**
8:         $h \leftarrow \operatorname{argmin}_{i \in \mathcal{N}(j)} D(\bar{x}, x_i)$, where ties are broken to prefer nodes with the lowest id.[6]
9:         **if** $D(\bar{x}, x_h) < D(\bar{x}, x_j)$ **then**
10:            $j \leftarrow h$.
11:         **else if** $D(\bar{x}, x_h) = D(\bar{x}, x_j)$ and $h < j$ **then**       ▷ Tie-breaking on node id.
12:            $j \leftarrow h$.
13:         **else**
14:            Done $\leftarrow$ True.
15: **Return** $x_j$

---

**Distance-Based Permutations.** We let $\mathcal{N}(i)$ denote the out-neighbors of node $i$ in the graph; i.e., $j \in \mathcal{N}(i)$ if and only if $(i, j) \in E$. We use the notation $N_1(i), \ldots, N_n(i)$ to index the list of nodes in the graph ordered in non-decreasing order by their distances from $i$; i.e., for $k < \ell$, $D(x_i, x_{N_k(i)}) \leq D(x_i, x_{N_\ell(i)})$. Ties are broken by node id. Specifically, whenever $D(x_i, x_{N_k(i)}) = D(x_i, x_{N_\ell(i)})$ and $k < \ell$, then $N_k(i) < N_\ell(i)$. This choice is arbitrary, but a consistent way of breaking ties will simplify our exposition. For most applications, we will not have distances repeat *exactly* in the dataset, so tie breaking is never invoked. Note that since we assume $x_1, \ldots, x_n$ are distinct and that $D(x_i, x_i) = 0$ for all $i$ and $D(x_i, x_j) > 0$ for all $j \neq i$, we always have that $N_1(i) = i$.

**Greedy Search and Navigability.** We study a notion of navigability that is tied to the standard greedy graph search algorithm for nearest neighbors, which is detailed in Algorithm 1. It can be checked that the algorithm always terminates in at most $n$ iterations since $j$ can only be equal to every node in the graph at most once. Additionally, we note that Line 11 and 12 are unnecessary in the case when distances are assumed to be unique. When this is not the case, these lines implement our arbitrary tie-breaking rule, which is to prefer nodes with lower id when distances are equal.

Given Algorithm 1, we define navigability formally as follows:

**Definition 3** (Navigable Graph). *A graph $G$ is* navigable *for point set $x_1, \ldots, x_n$ under distance function $D$ if, for all $s, t \in \{1, \ldots, n\}$, when Algorithm 1 is run with starting node $s$ and query $\bar{x} = x_t$, then the algorithm returns $x_t$. I.e., when the query $\bar{x}$ exactly matches a point in the dataset, the algorithm finds that point. We further say that $G$ is "small world" with parameter $S$ if the algorithm always terminates after at most $S$ calls to the while loop.*

It will be useful to think about navigability as a property of the distance-based node permutations defined earlier. In particular, navigability is *implied* by the following property:

$$\text{For all } t \text{ and all } \ell > 1, \text{ there is an edge from } N_\ell(t) \text{ to } N_k(t) \text{ for at least one } k < \ell. \qquad (1)$$

In particular, when given a query $x_t \in \{x_1, \ldots, x_n\}$, Algorithm 1 will only move from nodes $N_\ell(t)$ to $N_k(t)$ for which $k < \ell$. Moreover, as long as there is such a $k$ in the out-neighborhood of $N_\ell(t)$, then the algorithm will not terminate at $N_\ell(t)$. It follows that, if (1) holds, the algorithm is guaranteed to terminate at $N_1(t) = t$ and return $x_t$, as desired. We remark that, if all distances between nodes are distinct, (1) is equivalent to the navigability property of Definition 3, although we will not require this fact. To better illustrate the connection between (1) and Definition 3, we include an example of a navigable graph in Figure 1 and the corresponding list of distance-based permutations in Figure 2.

---

[6]Formally, $h \leftarrow \min\left(\{\operatorname{argmin}_{i \in \mathcal{N}(j)} D(\bar{x}, x_i)\}\right)$, where $\operatorname{argmin}_t f(t)$ returns the set of minimizers of $f$.

# 3 Upper Bound

We begin by proving our main positive result, which we restate below:

**Theorem 1.** *For any input domain $\mathcal{X}$, point set $x_1, \ldots, x_n \in \mathcal{X}$, and distance function $D : \mathcal{X} \times \mathcal{X} \to \mathbb{R}^{\geq 0}$ such that $D(x_i, x_i) = 0$ for all $i$ and $D(x_i, x_j) > 0$ for $x_j \neq x_i$, it is possible to efficiently construct a directed navigable graph with average degree at most $2\sqrt{n \ln n}$. Moreover, the graph has the additional "small world" property: greedy routing always succeeds in at most 2 steps.*

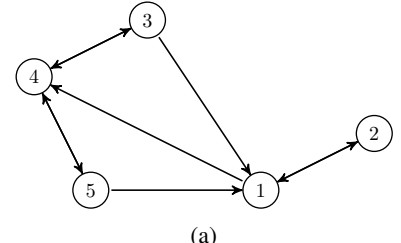

| Vertex | Coordinates |
|--------|-------------|
| 1 | $(4, -2)$ |
| 2 | $(6, -1)$ |
| 3 | $(2, 1)$ |
| 4 | $(0, 0)$ |
| 5 | $(1, -2)$ |

(a)                                  (b)

Figure 1: Example of a navigable graph $G = (V, E)$ on 5 nodes. A double arrow indicates that both $(i, j) \in E$ and $(j, i) \in E$. We can check that $G$ is navigable by referring to Figure 2.

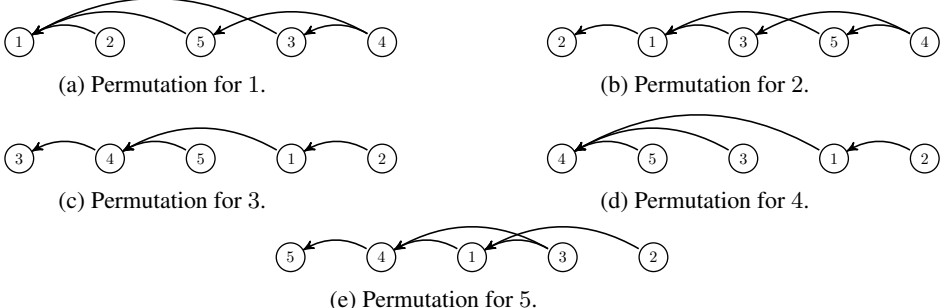

Figure 2: Distance-based permutations for the data set in Figure 1. As an example, in the plot above, we have $N_1(1) = 1, N_2(1) = 2, N_3(1) = 5, N_4(1) = 4, N_5(1) = 3$, we have $N_1(2) = 2, N_2(2) = 1, N_3(2) = 3, N_4(2) = 5, N_5(2) = 4$, etc. For the permutations, we draw all edges in the graph $G$ from Figure 1 that point "left" in the permutation. In particular, we show edges that connect any $N_\ell(t)$ to $N_k(t)$ with $k < \ell$. We can check that property (1) holds, so the graph is navigable.

*Proof.* We give two different constructions that establish the theorem. The first is randomized, and succeeds with high probability. The second is deterministic. Both require $O(n^2(T + \log n))$ time to construct, where $T$ is the time to evaluate the distance function $D$ for any two input points.

**Construction 1: Randomized.** Let $m$ be an integer between $1$ and $n$, to be chosen later. Our first construction is as follows:

- For all $i \in \{1, \ldots, n\}$ and all $1 < \ell \leq m$, add an edge from $N_\ell(i)$ to $N_1(i) = i$.

- For all $i \in \{1, \ldots, n\}$, add an edge from $i$ to $\lceil \frac{3n \ln n}{m} \rceil$ nodes chosen uniformly at random from $\{1, \ldots, n\} \setminus \{i\}$.

To prove that this construction leads to a navigable graph, we need to prove the (1) holds with high probability. To do so, consider the permutation $N_1(i), \ldots, N_n(i)$ for a fixed node $i$. The property trivially holds for all $\ell \leq m$ since we connected $N_\ell(i)$ to $N_1(i)$ in step one of the construction. So, we only have to consider $\ell > m$.

For any $\ell > m$, the chance that any one random edge from the second step of the construction connects to some $N_k(i)$ for $k \leq m$ is $\frac{m}{n}$. So, the chance that *none of the random edges* connect $N_\ell(i)$ to some $N_k(i)$ for $k \leq m$ is at most $\left(1 - \frac{m}{n}\right)^{\lceil \frac{3n \ln n}{m} \rceil} \leq \frac{1}{e^{3 \ln n}} \leq \frac{1}{n^3}$. By a union bound, it follows

that with probability at least $1 - \frac{1}{n}$, for all $i$ and all $\ell > m$, $N_\ell(i)$ has an edge to some $N_k(i)$ with $k \le m < \ell$. Thus property (1) holds, and we conclude that the graph we constructed is navigable.

It is left to set $m$. The graph we constructed has at most $(m-1)n + n \cdot \lceil \frac{3n \ln n}{m} \rceil \le mn + n \cdot \frac{3n \ln n}{m}$ edges. Balancing terms, if we choose $m = \sqrt{3n \ln n}$, the graph has $\le 2\sqrt{3} n^{1.5} \sqrt{\ln n}$ edges.

Finally, we observe that constructing the graph requires computing and sorting all $n$ distance-based permutations, which takes $O(n^2(T + \log n))$ time. Moreover, we can see that the constructed graph is small-world with parameter $C = 2$. In the first iteration of Algorithm 1, we are guaranteed to choose a node $h$ that is one of the $m$ closest neighbors of the input $\bar{x} = x_i$. Then, at the second iteration, $h$ has an edge to $i$ itself and so we terminate.

**Construction 2: Deterministic.** Our randomized construction can be derandomized relatively directly. We construct the same $n(m-1)$ edges from $N_\ell(i)$ to $N_1(i)$ for all $i$ and $1 < \ell \le m$. The goal in constructing random edges was to ensure that, for $\ell > m$, $N_\ell(i)$ always has an edge to some node in $\{N_1(i), \dots, N_m(i)\}$, which we will call $i$'s "near-neighborhood", and denote by $\mathcal{N}_m(i)$. We can instead ensure that each $N_\ell(i)$ has an edge into $\mathcal{N}_m(i)$ via a greedy set cover approach.

In particular, we claim that there is a set of $g \le 1 + \frac{n \ln n}{m}$ nodes $k_1, \dots, k_g$ such that every near-neighborhood $\mathcal{N}_m(i)$ contains at least one of $k_1, \dots, k_g$. To ensure property (1) for values of $\ell > m$, we only have to connect all nodes in our graph to this set. We construct this set greedily. Let $B_1 = \{1, \dots, n\}$ and, for $i > 1$, let $B_i$ denote the set of all $i$ for which none of $k_1, \dots, k_{i-1}$ is in $\mathcal{N}_m(i)$. We have that $|B_1| = n$ and our goal is to show that $|B_{g+1}| < 1$. By a counting argument, there must be at least one node that appears in $\mathcal{N}_m(i)$ for at least $m$ different values of $i \in B_1$. Select this node to be $k_1$, which ensures that:

$$|B_2| \le \left(1 - \frac{m}{n}\right) |B_1| = \left(1 - \frac{m}{n}\right) n.$$

Again by a counting argument, there must be one node that appears in $\mathcal{N}_m(i)$ for at least $\frac{|B_2| \cdot m}{n}$ values of $i \in B_1$. Select this node to be $k_2$, which ensures that:

$$|B_3| \le |B_2| - \frac{|B_2| \cdot m}{n} = \left(1 - \frac{m}{n}\right) |B_2| \le \left(1 - \frac{m}{n}\right)^2 n.$$

Continuing in this way, we conclude that $|B_{g+1}| \le \left(1 - \frac{m}{n}\right)^g n$, which is less than 1 as long as $g > \frac{n \ln n}{m}$. We conclude that, as long as we connect every node to $k_1, \dots, k_g$, (1) is satisfied for all $N_\ell(i)$ where $\ell > m$, so our constructed graph is navigable.

In total, our deterministic graph has at most $(m-1)n + n \cdot \left(\frac{n}{m} \ln n + 1\right)$ edges. Choosing $m = \sqrt{n \ln n}$ we get a graph with at most $2n^{1.5} \sqrt{\ln n}$ edges. Again, the cost of the algorithm is dominated by the $O(n^2(T + \log n))$ time required to compute and sort all $n$ distance-based permutations. $\square$

We remark that, while it may be possible to improve our upper bound from $O(\sqrt{n \log n})$ to $O(\sqrt{n})$ average degree, we do not believe our analysis can be directly tightened. For random points in high-dimensional space under the Euclidean metric, we roughly expect each near-neighborhood $\mathcal{N}_m(i)$ to look like a uniformly random subset of $\{1, \dots, n\}$. If the neighborhoods were truly random, then existing results on random set cover problems imply that it is not possible to find $k_1, \dots, k_g$ covering all near-neighborhoods unless $g = \Omega\left(\frac{n \ln n}{m}\right)$ [Vercellis, 1984, Arpino et al., 2024].

## 4 Lower Bounds

In this section, we prove Theorem 2, which shows that even for a random point set in $\mathbb{R}^d$ for $d = O(\log n)$ under the Euclidean metric, Theorem 1 cannot be improved significantly: with high probability, any navigable graph $G$ must have average degree $\Omega(n^{1/2-\delta})$ for any fixed constant $\delta$. Theorem 2 is a corollary of our more general Theorem 4, which we state below:

**Theorem 4.** *Let $x_1, \dots, x_n \in \{-1, 1\}^d$ be distributed independently and uniformly in $\{-1, 1\}^d$. With probability $> \frac{9}{10}$, any navigable graph for $x_1, \dots, x_n$ under the Euclidean metric requires $\Omega(n^{3/2-\epsilon})$ edges, for $\epsilon = \max\left(\frac{\log \log n}{\log n}, c\sqrt{\frac{\log n}{d}}\right)$ for a universal constant $c$.*

The $\Omega(n^{3/2-\delta})$ lower bound of Theorem 2 follows immediately from Theorem 4 by taking $d = \frac{c^2}{\delta^2}\log n$. Alternatively, if we take $d = c^2 \log^3 n$ we have a lower bound of $\Omega(n^{3/2}/\log n)$, which matches Theorem 1 up to a $O(\log^{3/2} n)$ factor.

## 4.1 Average Degree Lower Bound

We introduce a few intermediate results before giving the proof of Theorem 4. First, we observe that the point set of Theorem 4 does not have any duplicate points with high probability. Doing so simplifies our analysis as no "tie-breaking" will be needed when Algorithm 1 is run on an input $x_j$.

**Claim 5** (No Repeated Points). *Let $x_1, \ldots, x_n$ be distributed as in Theorem 4. As long as $d \geq c_1 \log n$ for a universal constant $c_1$, then with probability at least $99/100$, all vectors in this set are distinct.*

The claim follows from a simple probability calculation and union bound – see Appendix A.

We next define a notion of a neighborhood of a point, which includes all points within a certain radius.

**Definition 6** (Fixed Radius Near-Neighborhood). *Consider the setting of Theorem 4. Let $\mathcal{O}_j$ be the subset of vectors in $x_1, \ldots, x_n$ (including $x_j$ itself) with $\langle x_i, x_j \rangle \geq c_h \sqrt{d \log n}$ where $c_h \in [1/3, 1]$ is some value (that may depend on $n$) which we will specify later.*

Note that, for any $x_i, x_j \in \{-1, 1\}^d$, $\|x_i - x_j\|_2^2 = d - 2\langle x_i, x_j \rangle$, so $\mathcal{O}_j$ contains a set of nearest neighbors to $j$ in the Euclidean distance. Importantly, however, the definition used in this section is different from the $\mathcal{N}_m(j)$ notation used in the previous section, since $\mathcal{O}_j$ is not of a fixed size.

Using the distance-based permutation characterization of navigability given in (1), we can observe that, in order for greedy routing to make progress towards target $x_j$, every node in the near-neighborhood $\mathcal{O}_j$ needs an edge to another node in the near-neighborhood, closer to $x_j$. Formally:

**Claim 7** (Required Connections Within Neighborhoods). *Consider the setting of Theorem 4 and assume that $x_1, \ldots, x_n \in \{-1, 1\}^d$ are distinct. Any navigable graph $G$ for $x_1, \ldots, x_n$ requires $|\mathcal{O}_j| - 1$ edges in $\mathcal{O}_j \times \mathcal{O}_j$ for each $j$.*

*Proof.* For $G$ to be navigable, for any $i \in \mathcal{O}_j \setminus \{j\}$, there must be an edge from $i$ to some node that is closer $j$, and thus is also in $\mathcal{O}_j$. Thus, there are at least $|\mathcal{O}_j| - 1$ edges in $\mathcal{O}_j \times \mathcal{O}_j$. $\square$

We next make two claims about the near neighborhoods of Definition 6 when $x_1, \ldots, x_n$ are random points in $\{-1, 1\}^d$: that 1) they are large with high probability and 2) that they have low overlap with high probability. Together with Claim 7, these imply that any navigable graph $G$ for $x_1, \ldots, x_n$ requires a large number of edges, proving Theorem 4. We first formally state the claims and prove Theorem 4 using them. We then prove the claims in Section 4.2.

**Claim 8** (Neighborhoods are Large). *Let $x_1, \ldots, x_n$ be as distributed as in Theorem 4 and let $\mathcal{O}_j$ be as in Definition 6. As long as $d \geq c_1 \log n$ for a universal constant $c_1$, with probability at least $99/100$, for all $j$, $|\mathcal{O}_j| \geq \sqrt{n}/6$.*

**Claim 9** (Neighborhood Intersections are Small). *Let $x_1, \ldots, x_n$ be distributed as in Theorem 4 and let $\mathcal{O}_j$ be as in Definition 6. As long as $d \geq c_1 \log n$ for a universal constant $c_1$, with probability at least $99/100$, for all $i \neq j$, $|\mathcal{O}_i \cap \mathcal{O}_j| \leq 10 \max\left(\log n, n^{c\sqrt{\frac{\log n}{d}}}\right)$ for some universal constant $c$.*

Claim 8 establishes that $\mathcal{O}_j$ contains the $\Theta(\sqrt{n})$ nearest neighbors to $x_j$, which is a consequence of our choice of radius in Def. 6. If each $\mathcal{O}_j$ were just an independent random set of $\Theta(\sqrt{n})$ nodes, then we would expect that for any $i \neq j$, $|\mathcal{O}_i \cap \mathcal{O}_j| = \Theta(1)$. I.e., our neighborhoods would have small overlap, which is the key property we need to prove a lower bound. An overlap of $O(1)$ would imply that $O(n^{3/2})$ edges are needed to satisfy the requirements of Claim 7. Of course, each $\mathcal{O}_j$ is not an independent random set in reality. Specifically, if $x_i$ and $x_j$ have large inner product, $\mathcal{O}_i$ and $\mathcal{O}_j$ will be correlated, so we expect that $|\mathcal{O}_i \cap \mathcal{O}_j|$ will be larger. Claim 9 shows that for large enough $d$, such strong correlations are unlikely to happen and thus we still expect $|\mathcal{O}_i \cap \mathcal{O}_j|$ to be fairly small.

***Proof of Theorem 4.*** First note that we can assume that $d \geq c_1 \log n$ for some large constant $c_1$, as otherwise we will have $\epsilon > 3/2$ and the lower bound becomes vacuous. Accordingly, the conclusions of Claims 5, 8 and 9 all hold for $x_1, \ldots, x_n$ with probability $> 9/10$ by a union bound.

We will use these claims to show that any navigable graph for $x_1, \ldots, x_n$ requires $\Omega(n^{3/2-\epsilon})$ edges. Consider any navigable graph $G = (V, E)$. For any edge $(u, v) \in E$, let $w_{u,v} = |\{j : u, v \in \mathcal{O}_j\}|$ be the number of near neighborhoods that $u$ and $v$ both belong to. I.e., $w_{u,v}$ is the number of nodes $j$ for which $(u, v) \in \mathcal{O}_j \times \mathcal{O}_j$. By Claims 7 and 8, we must have

$$\sum_{(u,v) \in E} w_{u,v} \geq \sum_{j=1}^{n} |\mathcal{O}_j| - 1 \geq \frac{n^{3/2}}{6} - n \geq \frac{n^{3/2}}{12}, \tag{2}$$

where we use in the last step that $n \leq n^{3/2}/12$ for large enough $n$.

Further, $w_{u,v} = |\mathcal{O}_u \cap \mathcal{O}_v|$ since $u$ and $v$ both lie in $\mathcal{O}_j$ exactly when $j$ lies in both $O_u$ and $\mathcal{O}_v$. Thus, by Claim 9, $w_{u,v} \leq 10 \max(\log n, n^{c\sqrt{\log n/d}})$ for all $u, v$. Combined with (2) this gives:

$$|E| \geq \frac{n^{3/2}/12}{10 \max(\log n, n^{c\sqrt{\log n/d}})} = \Omega(n^{3/2-\epsilon}),$$

for $\epsilon = \max(\frac{\log \log n}{\log n}, c\sqrt{\log n/d})$. This proves the theorem. $\square$

## 4.2 Probabilistic Claims about Near Neighborhoods

We now prove Claims 8 and 9. We will use a very sharp bound on the CDF of a binomial distribution, given by Ahle [2017] and attributed to Cramer [Cramér, 2022]. This is a quantitative version of the central limit theorem, saying that the binomial CDF is close to the normal CDF, up to some small error. It is tighter than more general bounds like the Berry-Esseen theorem. We give a proof of our exact statement of the bound in Appendix A.

**Fact 10** (Binomial CDF Bound, 2.20 of [Ahle, 2017]). *Let $F_t(\cdot)$ be the CDF of a mean centered binomial random variable with $t$ trials and success probability $1/2$. There are universal constants $c_1, c_2$ such that, for any $x$ satisfying $c_1 \leq x \leq \sqrt{t}/c_1$,*

$$\frac{1}{3x} e^{-\frac{x^2}{2} \cdot \left(1 + \frac{c_2 x^2}{t}\right)} \leq F_t(-x \cdot \sqrt{t}/2) \leq \frac{1}{x} e^{-\frac{x^2}{2} \cdot \left(1 - \frac{c_2 x^2}{t}\right)}.$$

We will also use that, with high probability, no pair of our random vectors has too high of an inner product. This will be required to prove Claim 9, i.e., that all near neighborhoods have small overlap. The claim follows directly from a standard Chernoff bound and a union bound over all pairs $i, j$.

**Claim 11** (Vectors are Not Too Similar). *Let $x_1, \ldots, x_n$ be distributed as in Theorem 4. As long as $d \geq c_1 \log n$ for a universal constant $c_1$, with probability at least $99/100$, for all $i \neq j$, $\langle x_i, x_j \rangle \leq c_u \sqrt{d \log n}$ for some fixed constant $c_u$.*

Finally, we require the following technical claim, which we will use to set $c_h$ in Definition 6. Our goal is to ensure that the probability of a vector being in the near neighborhood of another is $\Theta(1/\sqrt{n})$.

**Claim 12.** *For any large enough $n$, there is some value of $c \in [1/3, 1]$ such that*

$$\frac{1}{\sqrt{\ln n}} \cdot \exp(-c^2 \cdot \ln n) = \frac{1}{\sqrt{n}}.$$

*Proof.* Observe that if we set $c = 1$, then

$$\frac{1}{\sqrt{\ln n}} \cdot \exp(-c^2 \cdot \ln n) = \frac{1}{n\sqrt{\ln n}} < \frac{1}{\sqrt{n}}.$$

Further, if we set $c = 1/3$, we have for large enough $n$,

$$\frac{1}{\sqrt{\ln n}} \cdot \exp(-c^2 \cdot \ln n) = \frac{1}{\sqrt{\ln n}} \exp(-1/9 \ln n) = \frac{1}{\sqrt{\ln n} \cdot n^{1/9}} > \frac{1}{\sqrt{n}}.$$

Thus, for some setting of $c \in [1/3, 1]$ the claim holds. $\square$

***Proof of Claim 8.*** Fix some $x_j$. Then for any $i \neq j$, the event that $i \in \mathcal{O}_j$ is exactly the event that a binomial random variable $Bin(d, 1/2)$ exceeds its mean by $\geq c_h\sqrt{d \ln n}/2$. I.e., letting $F_d(\cdot)$ be the CDF of the mean centered binomial random variable, we have

$$\Pr(x_i \in \mathcal{O}_j) = F_d(-c_h\sqrt{d \ln n}/2). \tag{3}$$

Plugging in the lower bound of Fact 10, we have for some constant $c_2$:

$$\Pr(x_i \in \mathcal{O}_j) \geq \frac{1}{3 \cdot c_h\sqrt{\ln n}} \cdot \exp\left(-\frac{c_h^2}{2} \cdot \ln n \cdot \left(1 + \frac{c_2 c_h^2 \ln n}{d}\right)\right).$$

Using that $d \geq c_1 \ln n$ for some sufficiently large constant $c_1$, $1 + \frac{c_2 c_h^2 \ln n}{d} \in [1, 2]$. Then, by Claim 12, we can set $c_h$ to some value in $[1/3, 1]$ to obtain:

$$\Pr(x_i \in \mathcal{O}_j) \geq \frac{1}{3c_h\sqrt{n}} \geq \frac{1}{3\sqrt{n}}. \tag{4}$$

From (4), the claim follows by noting that the events $i \in \mathcal{O}_j$ are independent and each happens with probability at least $\frac{1}{3\sqrt{n}}$, so $\mathbb{E}\left[|\mathcal{O}_j|\right] \geq \sqrt{n}/3$. Thus, by a Chernoff bound, with probability at least $1 - 2^{\Omega(\sqrt{n})}$, $|\mathcal{O}_j| \geq \sqrt{n}/6$. Taking a union bound over all $j$ then gives that, for sufficiently large $n$, $|\mathcal{O}_j| \geq \sqrt{n}/6$ for all $j$ with probability at least $99/100$, giving the claim. $\qquad\square$

***Proof of Claim 9.*** Fix $i \neq j$ and consider some $k$ which is not equal to either $i$ or $j$. Let $z$ be the number of positions in which $x_i$ and $x_j$ take the same value. Note that, assuming the event of Claim 11 holds, $z \leq d/2 + c_u\sqrt{d \log n}/2$. Denote $d/2 + c_u\sqrt{d \log n}/2$ by $\mu$.

By Definition 6, to have $x_k \in \mathcal{O}_i \cap \mathcal{O}_j$ we need $\min(\langle x_k, x_i\rangle, \langle x_k, x_j\rangle) \geq c_h\sqrt{d \log n}$. That is, we need $x_k$ to match each of $x_i$ and $x_j$ in at least $d/2 + c_h/2 \cdot \sqrt{d \log n}$ positions. On the $d - z$ positions where $x_i$ and $x_j$ differ, the best case scenario (i.e., the scenario maximizing $\min(\langle x_k, x_i\rangle, \langle x_k, x_j\rangle)$) is when $x_k$ matches exactly half of these positions for each of $x_i, x_j$ (i.e., matches $\frac{d-z}{2}$ positions).

Assuming this case, to have $x_k \in \mathcal{O}_i \cap \mathcal{O}_j$, on the $z$ positions where $x_i$ and $x_j$ are identical, $x_k$ must match $z/2 + c_h/2 \cdot \sqrt{d \log n}$ positions. The probability of this happening is equivalent to the probability that a binomial random variable with $z$ trials exceeds its mean by $\geq c_h/2 \cdot \sqrt{d \log n}$. Since $z \leq \mu$, we can upper bound this probability by the probability that a binomial random variable with $\mu$ trials exceeds its mean by $c_h/2 \cdot \sqrt{d \log n}$. We rewrite the upper bound as:

$$c_h/2 \cdot \sqrt{d \log n} = c_h/2 \cdot \sqrt{d/\mu} \cdot \sqrt{\mu \log n} = \frac{c_h}{\sqrt{2}} \cdot \sqrt{\frac{1}{1 + c_u\sqrt{\frac{\log n}{d}}}} \cdot \sqrt{\mu \log n},$$

where we use $\frac{d/2}{\mu} = \frac{1}{1 + c_u\sqrt{\frac{\log n}{d}}}$. Via Fact 10, we now upper bound our probability of interest by:

$$\Pr(x_k \in \mathcal{O}_i \cap \mathcal{O}_j) \leq \frac{\sqrt{1 + c_u\sqrt{\frac{\log n}{d}}}}{\sqrt{2}c_h\sqrt{\log n}} \cdot \exp\left(-\frac{c_h^2 \log n}{1 + c_u\sqrt{\frac{\log n}{d}}} \cdot \left(1 - \frac{2c_2 c_h^2 \cdot \log n}{(1 + c_u\sqrt{\frac{\log n}{d}}) \cdot \mu}\right)\right).$$

Observe that since in the claim we require $d \geq c_1 \log n$ for a large constant $c_1$, we have that $1 \leq 1 + c_u\sqrt{\frac{\log n}{d}} \leq 2$. So, we can simplify the above to:

$$\Pr(x_k \in \mathcal{O}_i \cap \mathcal{O}_j) \leq \frac{\sqrt{2}}{c_h\sqrt{\log n}} \cdot \exp\left(-\frac{c_h^2 \log n}{1 + c_u\sqrt{\frac{\log n}{d}}} \cdot \left(1 - \frac{2c_2 c_h^2 \cdot \log n}{\mu}\right)\right)$$

$$= \frac{\sqrt{2}}{c_h\sqrt{\log n}} \cdot \exp\left(-c_h^2 \log n \cdot \left(1 + \frac{c_2 c_h^2 \log n}{d}\right) \cdot \frac{\left(1 - \frac{2c_2 c_h^2 \cdot \log n}{\mu}\right)}{\left(1 + \frac{c_2 c_h^2 \log n}{d}\right) \cdot \left(1 + c_u\sqrt{\frac{\log n}{d}}\right)}\right).$$

Now, since we set $d \geq c_1 \log n$ for a large constant $c_1$, we can ensure that have that $\frac{\log n}{d} \leq c_3 \frac{\sqrt{\log}}{\sqrt{d}}$ for any small constant $c_3$. Using this fact, along with the fact that $\mu \geq d/2$, we have:

$$\frac{\left(1 - \frac{2c_2 c_h^2 \cdot \log n}{\mu}\right)}{\left(1 + \frac{c_2 c_h^2 \log n}{d}\right) \cdot \left(1 + c_u \sqrt{\frac{\log n}{d}}\right)} \geq \frac{\left(1 - .5 c_u \sqrt{\frac{\log n}{d}}\right)}{\left(1 + .5 c_u \sqrt{\frac{\log n}{d}}\right) \cdot \left(1 + c_u \sqrt{\frac{\log n}{d}}\right)} \geq 1 - 2c_u \cdot \sqrt{\frac{\log n}{d}}.$$

For the second inequality, we used the fact that $\frac{(1-.5x)}{(1+.5x)(1+x)} \geq 1 - 2x$ for all $x$. Next, setting $c_h \in [1/3, 1]$ exactly as in (4) so that $\frac{1}{\sqrt{\log n}} \cdot \exp\left(-\frac{c_h^2}{2} \log n \cdot \left(1 + \frac{c_2 c_h^2 \log n}{d}\right)\right) = \frac{1}{\sqrt{n}}$, we have:

$$\Pr(x_k \in \mathcal{O}_i \cap \mathcal{O}_j) \leq \frac{\sqrt{2}}{c_h \sqrt{\log n}} \cdot \exp\left(-c_h^2 \log n \cdot \left(1 + \frac{c_2 c_h^2 \log n}{d}\right) \cdot \left(1 - 2c_u \sqrt{\frac{\log n}{d}}\right)\right)$$

$$\leq \frac{\sqrt{2}}{c_h n} \cdot \exp\left(2c_h^2 c_u \log n \cdot \left(1 + \frac{c_2 c_h^2 \log n}{d}\right) \cdot \sqrt{\frac{\log n}{d}}\right)$$

$$\leq \frac{4\sqrt{2}}{n} \cdot \exp\left(\frac{c \log^{3/2} n}{d}\right) = \frac{4\sqrt{2}}{n} \cdot n^{c\sqrt{\frac{\log n}{d}}},$$

for some constant $c$. Finally, observe that, conditioned on $x_i$ and $x_j$ sharing $z$ entries, the event $x_k \in \mathcal{O}_i \cap \mathcal{O}_j$ is independent for each $x_k$. Thus, by a standard Chernoff bound, with probability at least $1 - 1/n^{c'}$, we have $|\mathcal{O}_i \cap \mathcal{O}_j| \leq 10 \max(\log n, n^{c\sqrt{\frac{\log n}{d}}})$ for some large constant $c'$. Union bounding over all $O(n^2)$ pairs $i \neq j$ then gives the claim. $\qquad\square$

### 4.3 Maximum Degree Lower Bound

In Theorems 1 and 2, we focus on the *average degree* of navigable graphs. While a reasonable metric, we might also be interested in the *maximum degree*, which governs the worst-case complexity of a single step of the greedy algorithms. Here we give a simple argument showing that unfortunately, on a worst-case input instance, it is not possible to achieve maximum degree better than the trivial $n - 1$ given by the complete graph.

**Theorem 13.** *There exists a set of points $x_1, \ldots, x_n \in \mathbb{R}^d$ for $d = O(\log n)$ such that any navigable graph for these points under the Euclidean metric has maximum out-degree $d - 1$.*

*Proof.* First we prove the bound for a high dimensional point set. For all $i < n$, let $x_i$ be a standard basis vector with a 1 in position $i$. Let $x_n$ be the all zeros vector. As we can see, $x_n$ has Euclidean distance 1 from $x_1, \ldots, x_{n-1}$. On the other hand, for all $i \neq j$ where $i, j \neq n$, $\|x_i - x\|_2 = \sqrt{2}$. In other words, $x_n$ is the closest neighbor to each of $x_1, \ldots, x_{n-1}$. It follows that any navigable graph *must* contain an edge from $x_1$ to each of these points, resulting in maximum degree $n$.

To extend the construction to low dimensions, we simply use the Johnson-Lindenstrauss Lemma to embed the point set above into $c \log n$ dimensions. As long as the constant $c$ is sufficiently large, all distances will be preserved to within error, say, $\pm 0.1$. So, $x_n$ will still be the nearest neighbor of all other points, and we will still require it to have out-degree $n - 1$. $\qquad\square$

## Acknowledgements

Christopher Musco was supported in part by NSF award IIS-2106888. Cameron Musco was supported in part by NSF award CCF-2046235 and an Adobe Research grant.

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

# A Omitted Proofs

Below we gives the proofs of some simple technical claims required to prove our main lower bound, Theorem 4. First, we prove Claim 5, which establishes that, with high probability, the hard input distribution of Theorem 4 produces a set of distinct points..

**Claim 5** (No Repeated Points). *Let $x_1, \ldots, x_n$ be distributed as in Theorem 4. As long as $d \geq c_1 \log n$ for a universal constant $c_1$, then with probability at least $99/100$, all vectors in this set are distinct.*

*Proof.* Consider two points $x_i, x_j$. The probability that these points are identical equals $\frac{1}{2}^d \leq \frac{1}{n^{c_1}}$. Then by a union bound over all pairs $i, j$, we have that all points are distinct with probability at least $1 - \frac{1}{n^{c_1-2}}$, which is greater than $99/100$ for sufficiently large $c_1$. $\qquad\square$

Next we give a short derivation of Fact 10, which gives a sharp bound on the CDF of a binomial distribution, and is used in proving Claims 8 and 9.

**Fact 10** (Binomial CDF Bound, 2.20 of [Ahle, 2017]). *Let $F_t(\cdot)$ be the CDF of a mean centered binomial random variable with $t$ trials and success probability $1/2$. There are universal constants $c_1, c_2$ such that, for any $x$ satisfying $c_1 \leq x \leq \sqrt{t}/c_1$,*

$$\frac{1}{3x} e^{-\frac{x^2}{2} \cdot \left(1 + \frac{c_2 x^2}{t}\right)} \leq F_t(-x \cdot \sqrt{t}/2) \leq \frac{1}{x} e^{-\frac{x^2}{2} \cdot \left(1 - \frac{c_2 x^2}{t}\right)}.$$

*Proof.* Applying 2.20 of [Ahle, 2017] with $p = q = 1/2$ gives that

$$F_t(-x \cdot \sqrt{t}/2) \in \frac{1}{\sqrt{2\pi}x} \exp\left(-tD\left(\frac{1}{2} - \frac{x}{2\sqrt{t}} \| \frac{1}{2}\right)\right) \cdot \left(1 \pm c\left(\frac{1}{x^2} + \frac{x}{\sqrt{t}}\right)\right),$$

for some constant $c$, where use the notation $D(a\|b) \overset{\text{def}}{=} a\log(a/b) + (1-a)\log((1-a)/(1-b))$. This is the KL divergence between two Bernoulli distributions with success probabilities $a$ and $b$. We can then apply equation (2.13) of [Ahle, 2017] to claim that, for a constant $c_2$,

$$D\left(\frac{1}{2} - \frac{x}{2\sqrt{t}} \| \frac{1}{2}\right) \in \frac{x^2}{2t} \pm c_2 \frac{x^4}{2t^2},$$

Plugging back in, we have that for constants $c$ and $c_2$,

$$F_t(-x \cdot \sqrt{t}/2) \in \frac{1}{\sqrt{2\pi}x} \exp\left(-\frac{x^2}{2} \cdot \left(1 \pm \frac{c_2 x^2}{t}\right)\right) \cdot \left(1 \pm c\left(\frac{1}{x^2} + \frac{x}{\sqrt{t}}\right)\right).$$

If we assume $c_1 \leq x \leq \sqrt{t}/c_1$ for large enough constant $c_1$ then $\frac{1}{3x} \leq \frac{1}{\sqrt{2\pi}x} \cdot \left(1 - c\left(\frac{1}{x^2} + \frac{x}{\sqrt{t}}\right)\right)$ and $\frac{1}{\sqrt{2\pi}x} \cdot \left(1 + c\left(\frac{1}{x^2} + \frac{x}{\sqrt{t}}\right)\right) \leq \frac{1}{x}$, which completes the proof. $\qquad\square$

