# OpenReview forum: "Navigable Graphs for High-Dimensional Nearest Neighbor Search: Constructions and Limits"
_NeurIPS.cc/2024/Conference — NeurIPS 2024 poster_

### Official Review · Reviewer_damy · 2024-07-09

**Soundness:** 4
**Presentation:** 4
**Contribution:** 3
**Rating:** 7
**Confidence:** 3

**Summary:**

The article establishes upper and lower bounds for the average degree of navigable graphs in high-dimensional case. In particular, a method for constructing a navigable graph with average degree $\mathcal{O}(\sqrt{n \log n})$ for any set of $n$ points is provided. In addition, they provide a random point set for which (with a high probability) it is not possible to build a navigable graph with average degree $\mathcal{O}(n^\alpha)$ for $\alpha < 1/2$.

**Strengths:**

Theoretical limits of the navigable graphs is an important question, since they are widely applied in state-of-the-art approximate nearest neighbor algorithms. The article provides sharp bounds for the average degree of the navigable graphs. The technical level of the article is high.

**Weaknesses:**

The article is motivated by the observation that the state-of-the-art methods for approximate nearest neighbor search utilize navigable graphs. However, as the authors acknowledge, these graph are in practice only approximately navigable, and use beam search to retrieve approximate nearest neighbors instead of a simple greedy search. Thus, the bounds provided are not directly applicable to these algorithms, but consider simplified version of the algorithms.

**Questions:**

If I understood correctly, you prove via a counterexample that it is not possible to build a navigable graph with an average degree smaller than $\mathcal{O}(\sqrt{n})$? for all possible sets of $n$ points? But this does not rule out that it would be possible to do it for a set of $n$ points for which certain additional assumptions holds?

**Limitations:**

The authors fairly address the limitations of the work.

---

> ### Author Rebuttal · Authors · 2024-08-06
>
> Thank you for the review. Regarding your question, your understanding is exactly correct. Some point sets admit sparser navigable graphs. For example, if all points lie on a $d$-dimensional hyperplane, then the Arya, Mount  result discussed in Section 1.1 implies that we can find a navigable graph with degree $2^{O(d)}$, which can be less than $O(\sqrt{n\log n})$ for small values of $d$. That said, our lower bound does not require an “adversarial” set of points: it holds with high-probability for a set of random $\pm 1$ vectors in $O(\log n)$ dimensions. Of course, “real data” often does not look random. An interesting question for future work might be to understand if there are natural measures of complexity of a point set that dictate the minimum degree of a navigable graph for that set.

---

> > ### Comment · Reviewer_damy · 2024-08-08
> >
> > I acknowledge that I have read the rebuttal, and thank the authors for carefully answering my question.

---

### Official Review · Reviewer_Jtod · 2024-07-10

**Soundness:** 3
**Presentation:** 2
**Contribution:** 2
**Rating:** 3
**Confidence:** 4

**Summary:**

This paper studies the problem of constructing navigable graphs over high-dimensional point sets. Specifically, a randomized algorithm and a deterministic algorithm are given to construct such graphs within almost the same time complexity. Besides, theoretical results demonstrate that both algorithms can achieve the average degree is O(\sqrt{nlogn}), which nearly matches the lower bound.

**Strengths:**

S1. The paper studies an important problem.
S2. Several interesting theoretical results are given
S3. The main ideas are easy to follow.

**Weaknesses:**

W1. There is no experimental evaluation.
W2. More related work should be reviewed.
W3. Some technical details require clarifications.
W4. A section of conclusion and future direction is missing.

**Questions:**

Q1. Navigable graphs have several applications now, but different applications may have different requirements on the efficiency, effectiveness, or their trade-off. However, the (potential) application scope of the proposed algorithms is unclear. Are they designed for exact NNS or approximate NNS? Are they designed for NNS only, or can it be extended to k-NNS?

Q2. Section 1.1 claims that “the computational efficiency of the graph-based methods … motivating the need for sparse navigable graphs”, which explains the main motivation to study this work. However, I found it unconvincing. First, some methods (eg HNSW and its variants) have already been demonstrated to be very efficient, so it might be meaningless for them to use the proposed algorithms. Second, the meaning of computational efficiency is a little vague. Take  HNSW as an example, its efficiency include two aspects: time efficiency for construction and time efficiency for query. Which one do you indicate here? Third, as mentioned in this paragraph, quite a few graph-based methods are essentially approximate solutions, so there will usually be a consideration on the trade-off between efficiency and effectiveness. Due to the reasons, I think the motivation will be more convincing, if the authors can provide more motivation studies here.

Q3. Two algorithms are proposed to tackle the same problem: a randomized one and a deterministic one. Moreover, their time complexity are both O(n^2(T+logn)), so it becomes meaningful to provide more discussions on their strengths and weaknesses.

Q4. The paper has no experimental evaluation, so it is hard to tell whether their proposed techniques can be helpful in practice.

Q5. The theoretical analysis mainly concentrates on the average degree, indicating the sparsity of graphs. There are other options to represent the sparsity, eg the maximum degree. Then, why selects average degree instead of other potential metrics? Please give more explanations.

Q6. Since the paper is related to the area of high-dimensional nearest neighbor search, there should more comprehensive literature review of recent studies on this topic, such as
[R1] RaBitQ: Quantizing High-Dimensional Vectors with a Theoretical Error Bound for Approximate Nearest Neighbor Search. SIGMOD 2024.
[R2] LiteHST: A Tree Embedding based Method for Similarity Search. SIGMOD 2023.
[R3] Turbo Scan: Fast Sequential Nearest Neighbor Search in High Dimensions. SISAP 2023.

Q7. The structure of this paper can be improved by summarizing the conclusion and identifying the future direction.

**Limitations:**

Please refer to the weaknesses and questions.

---

> ### Author Rebuttal · Authors · 2024-08-06
>
> Thank you for the detailed review. We provide responses to the specific questions below:
>
> Q1: There has been little formal work on connecting the property of navigability to near neighbor search. Indeed, the standard definition of navigability (which we study in our paper) only ensures that greedy search returns an exact NN for a query *in the dataset*. Navigability does not ensure greedy search works well for approximate NNS, k-nearest neighbor search, or exact nearest neighbor search for points not in the dataset. This important caveat is discussed in Section 1.3 of our paper.
>
> Nevertheless, we focus on navigability because it is often highlighted as a desirable feature in work on graph-based nearest neighbor search methods. For instance, navigability is essential for greedy search to obtain any bounded multiplicative approximation guarantee for approximate near-neighbor search (see our response to Reviewer Z5U8 for a detailed explanation of this fact). That said, we think an important research direction is to understand “generalized” notations of navigability that connect to the performance of approximate NN search, k-NN search, etc. For example, we are currently working on understanding notions of navigability that imply the convergence of the popular “beam search” method, a generalization of greedy search.
>
> Q2: We agree that graph-based methods have already been shown to be extremely efficient in practice. In fact, this is a major motivation for our work. Despite their empirical success, there is a lack of theoretical justification for this performance. Our paper aims to bridge this gap by enhancing the theoretical understanding of why graph-based methods are so effective, aligning with the goals of many related papers referenced in Section 1.3. We take a step in that direction by improving our theoretical understanding of navigability. We are not suggesting that our algorithms are ready to be used in practice in place of the effective graph-construction heuristics already used in methods like HNSW.
>
> In terms of computational efficiency, we were specifically referring to computational efficiency of the search, although we agree that the efficiency of graph construction (and efficient maintenance under dynamic updates) is important as well. We will clarify this in the paper.
>
> Q3: We will add further discussion of this point. The deterministic algorithm is “better” in that it has no chance of failure, and obtains a slightly smaller constant on the average degree. However, we chose to include the randomized algorithm because we think it is conceptually easier to understand, and aligns with prior strategies for constructing navigable graphs: we simply union together a near-neighbor graph and a random graph.
>
> Q4: As mentioned in our response to Q1, we do not recommend applying our methods in practical settings. The primary goal of this paper is to enhance theoretical understanding, which we believe may in turn result in further improvements to the approaches currently used in practice.
>
> Q5: This point is discussed further in Section 1.3 of the paper and Appendix B. In short, we agree that addressing other measures of sparsity is a great direction for future research. Maximum degree itself is a difficult one to work with, as there is a lower bound of $n-1$ (proven in Appendix B).
>
> Q6: Thank you for the suggested references!
>
> Q7: We summarize our contributions and elaborate on future directions in our “Outlook” section (Section 1.3). This could be moved to later in the paper to serve as a conclusion (possibly with some additional discussion).

---

> > ### Comment · Reviewer_Jtod · 2024-08-12
> > **Response to the author feedback**
> >
> > Dear authors,
> >
> > I have read the rebuttal, and thank you for considering my suggestions. I will carefully consider the rebuttal when making the final decision.
> >
> > Best regards,

---

### Official Review · Reviewer_MezS · 2024-07-12

**Soundness:** 3
**Presentation:** 4
**Contribution:** 3
**Rating:** 5
**Confidence:** 4

**Summary:**

This paper analyzes graph construction for greedy graph-based nearest neighbor search. First, the very general setup assuming an arbitrary similarity function is considered. In this case, it is shown that it is possible to construct a graph with an average degree at most $2 \sqrt{n \log n}$ which guarantees that the greedy search returns the nearest neighbor if the query coincides with an element of the dataset. Then, it is shown that the obtained average degree is close to the best possible since there are examples when the average degree cannot be better than $n^{\frac{1}{2}-\epsilon}$.

**Strengths:**

NNS is an important problem. Graph-based algorithms are widely used for this task and yet there is not much theoretical analysis of their performance. This paper addresses this gap and considers a very general setup where the dimension is essentially not limited (in other words - an arbitrary similarity function is given). While the main theoretical results are relatively simple (not much can be done in such a general setup), I enjoyed reading the paper. All results are clearly stated and motivated, the proofs are easy to follow. The related work is well described. Importantly, limitations are clearly mentioned in the text: the results assume that the query coincides with an element of the dataset, which is a significant limitation, as stressed in line 105.

**Weaknesses:**

Theorem 1 limits the average degree but does not say anything about the complexity of the search, which also depends on node degrees. It is shown in the paper that there are cases when the maximum degree in a graph is of order $n$, which means that the worst-case performance can be of order $n$ for this general setup. This can be a limitation of the considered general setup and some additional requirements can be needed to show something more feasible and practically applicable.


The considered general setup (that does not require the triangle inequality and is essentially based only on the ranking of neighbors) is similar to the one proposed in [1]. I think discussing this relation would improve the presentation. Also, [1] uses a relaxed triangle inequality to obtain better bounds (but for relatively small dimensions only).

[1] Goyal et al. "Disorder inequality: a combinatorial approach to nearest neighbor search." WSDM 2008.

Minor comments:
- Theorem 1 repeats two times in the main text, this repetition can be avoided: e.g., in the introduction only an informal description can be given.
- l263: "gives" should be "give"
- l391: $x$ should probably be $x_j$ here

**Questions:**

I do not have any questions in addition to the comments listed in the previous above.

**Limitations:**

Limitations are discusses in the paper.

---

> ### Author Rebuttal · Authors · 2024-08-06
>
> Thank you for the feedback. We address a few of the points raised below:
>
> -   We agree with the reviewer that an important next research direction is to look beyond the graph’s degree, and at more accurate proxies for the efficiency of near-neighbor search. For example, a natural metric might be the sum of degrees along any path taken by greedy search. It would be reasonable to study the average or worst case behavior of this metric for navigable graphs. We do feel that average degree serves as a good theoretical starting point, as it allows for direct comparison with prior work on low-dimensional data points, and is “simple”. We hope our progress on understanding average degree will lead to further theoretical progress on other graph metrics.
>
> -    Thanks very much for pointing out the Goyal et al.  reference, which we missed. Indeed, our general setup falls exactly into what they call “combinatorial near neighbor search”. We will add additional discussion to the final version of the paper.

---

> > ### Comment · Reviewer_MezS · 2024-08-11
> >
> > Thank you for the reply!
> >
> > While the results are relatively simple, it is a nice work that, I think, is above the acceptance threshold since it addresses an important problem (there is a gap between theory and practice in this field), provides new theoretical results and clearly states the limitations.

---

### Official Review · Reviewer_Z5U8 · 2024-07-12

**Soundness:** 3
**Presentation:** 2
**Contribution:** 3
**Rating:** 5
**Confidence:** 3

**Summary:**

The paper provides a theoretical framework for understanding and constructing navigable graphs for high-dimensional nearest neighbor search, and the authors establish some of the first upper and lower bounds for high-dimensional point sets.

**Strengths:**

S1: The article establishes both upper and lower bounds for navigable graphs and utilizes anti-concentration bounds for binomial random variables.

S2: The authors provide a foundational analysis that can influence the design and optimization of state-of-the-art ANNS methods, offering insights that could lead to more efficient graph-based solutions for high-dimensional data.

S3: The article uses a distance-based permutation method to analyze the lower and upper bounds of the navigable graphs, which is a highlight of the article.

**Weaknesses:**

W1: The method proposed in the article is not generalizable to the construction methods used by current graph-based approaches.

W2: The proposed graph construction algorithm has not been compared with existing graph construction algorithms. Such a method may only be useful for theoretical analysis rather than for building a practical application.

**Questions:**

Q1: Existing ANNS graph indexing algorithms do not necessarily need to ensure that the graph is fully navigable, as the query does not need to be considered a point in the graph. As long as the graph index can navigate to the vicinity of the given query vector, it largely meets the requirements. Therefore, the navigability of the graph is really important to ANNS?

Q2: It is better to make the proof or explanation more clear and detailed. For instance, providing more examples or describing the proof process in more formal language. This could be more helpful for someone who is not familiar with the theoretical research.

**Limitations:**

yes.

---

> ### Author Rebuttal · Authors · 2024-08-06
>
> Thank you for the thoughtful comments. We address the two specific questions below.
>
> Q1: A desirable property of an ANNS algorithm is that, if the vector queried, $q$, is in the dataset, then the algorithm should return exactly that vector. This behavior is also required for any algorithm to give a multiplicative approximation guarantee, i.e., any algorithm that always returns a vector $y$ satisfying $d(y,q) \leq \alpha\cdot \min_{i\in 1,\ldots, n} d(x_i,q)$ for $\alpha \geq 1$. In particular, if there is a point $x_i$ such that $d(x_i,q) = 0$, then for any finite $\alpha$, a multiplicative approximation algorithm must return that point.
>
> A major open research challenge is to prove strong multiplicative approximation guarantees for graph-based ANNS methods, similar to those available, e.g., for locality sensitive hashing. The argument above implies that navigability is *necessary* to achieve this goal. As the reviewer points out, it may also be reasonable to accept a weaker notion of approximation. However, we believe that the importance of multiplicative approximation in prior work on NNS supports the importance of navigability, and makes navigability a natural starting point to build on.
>
> More broadly, while existing graph-based near neighbor search methods like HNSW *usually* work fairly well in practice, they do fail on a subset of queries. One reason to explore navigability is to better understand these failures, and ultimately to make existing algorithms more reliable.
>
> Q2: Thank you for the feedback. We will keep this in mind when editing the paper. Please let us know if there are any specific proofs that could benefit from editing, or any examples that you recommend would help the reader further understand our work.

---

> > ### Comment · Reviewer_Z5U8 · 2024-08-14
> >
> > I acknowledge that I have read the rebuttal. Currently, Q2 has not been addressed clearly.

---

### Author Rebuttal · Authors · 2024-08-06

We would like to thank all of the reviewers for their thoughtful reviews and feedback. We address all specific questions in our individual responses below.

In general, we want to emphasize that our work can be viewed as a meaningful starting point to obtaining a better theoretical understanding of graph-based near neighbor search. Up until now, there has been very limited theoretical work on navigable graphs for high-dimensional data ($d > O(\log n)$). By making a first theoretical step on this topic, we hope to build a foundation for further theoretical work that can address many of the questions raised by the reviewers. For example, we plan to explore (and hope others will explore) different metrics of graph sparsity, the approximate search problem, generalizations of greedy search like beam search, and more.

---

### Decision · Program_Chairs · 2024-09-25

**Decision:**

Accept (poster)

**Comment:**

This paper aims to bridge our gap in theoretical understanding of very practical graph-based nearest neighbor search (NNS) methods in high dimensions. It provides a very clean theoretical view of the problem and proves almost matching lower and upper bounds on the size (average degree) of so-called navigable graphs for high-dimensional point sets.

There is no consensus among reviewers in the assessment of this paper. The main strengths they highlight are:
– Important problem.
– Tight bounds.
– Interesting techniques.
– Well written paper, easy to read.
The main weak points mentioned are:
– It is still quite far from explaining what the methods actually used in practice do.
– No impact on developing new practical solutions, no experimental evaluation.
– Bounds in terms of average degree and not max degree.
– Some related work not discussed.
– Some parts are hard to follow.

Based on the rebuttal, I believe the authors will discuss the missing related work in the next version of the paper. I also put little weight on the argument regarding experiments – I view this paper as a pure theory paper, whose goal is to increase our understanding of a certain phenomenon, and not to directly develop new practical algorithms.

The paper propose a very natural model to analyse graph-based NNS, and gives (almost) tight bounds in this model. It may seem somewhat disappointing and unimpressive that these bounds are far from what we observe in practice, but this only shows that this natural model is too simplistic, and we need to look for more elaborate ways to model the problem. It is an important first step that had to be made on our way to build our theoretical understanding of the power of graph-based NNS (in high dimension). Therefore I recommend to accept the paper. I believe it is going to inspire further notable work on this topic.